



**Title:** Comment on "Identification of the IMF sector structure in near-real time by ground magnetic
data" by Janzhura and Troshichev (2011).
**Author**: Peter Stauning, Danish Meteorological Institute, Lyngbyvej 100, Copenhagen.
Mail: pst@dmi.dk
**Abstract.** The only published description of the solar wind sector (SS) term used for the reference
level in the post-event and real-time derivation of the Polar Cap (PC) indices, PCN (North) and PCS
(South), in the version endorsed by the International Association for Geomagnetism and Aeronomy
(IAGA) is found in the commented publication, Janzhura and Troshichev: Identification of the IMF
sector structure in near-real time by ground magnetic data, Annales Geophysicae, 29, 1491-1500,
2011. Actually, the publication has served as basis for the index endorsement by IAGA in 2013.
However, neither the illustrations nor the results presented there have been derived by the specified
near-real time method. Figs. 1, 6, 7, and 8 display values derived by post-event calculations based
on daily medians smoothed over 7 days centred on the day of interest. Figs. 2, 3, and 4 display
observed values smoothed over 7 days, while the remaining Fig. 5 displays averages over 4 months.
In summary, there are strong disagreements between indications in the title, abstract, and statements
in the text compared to the actual results and their illustrations.
## 26   1. Introduction

The derivation of the Polar Cap (PC) indices, PCN (North) based on Qaanaaq data and PCS (South)
based on Vostok data, in the versions endorsed by the International Association for Geomagnetism
and Aeronomy (IAGA) in Resolution #3 (2013) is to a large extent based on the methods described
in Janzhura and Troshichev (2011): Identification of the IMF sector structure in near-real time by
ground magnetic data (hereinafter J&T2011) (and its replicate in Troshichev and Janzhura, 2012,
hereinafter T&J2012). This work provides the only published description of the solar wind sector
(SS) term related to the Y-component, IMF $B_Y$, of the interplanetary magnetic field (IMF). The SS
terms are derived from daily median values of the recorded magnetic field components and added to
the index reference level in the post-event or near-real time versions. For post-event PC index
calculations the SS-terms are derived from 7-days averaging of daily median values of the recorded
magnetic data. For the near-real time calculations the SS-terms are derived from cubic spline-based
forward extrapolation of past median values.
However, the method is invalid since it assumes that the IMF $B_Y$-related effects originating at the
dayside Cusp region can be compensated for by using a daily median-based SS-term at all local
hours. Instead, the addition of this singular term to the index reference level generates unfounded
positive or negative PC index contributions at different observatory positions along its daily path
with respect to the polar cap ionosphere. The solar wind sector "compensation", typically, generates
unfounded contributions to the PC indices at the night side although the real IMF $B_Y$ effects on
polar magnetic fields at the night side are usually very small. Correspondingly, the "compensation"
might have little effect on PC indices at the dayside although the Cusp-related IMF $B_Y$ effects
maximize there.

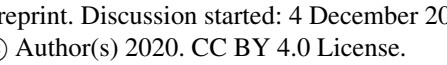

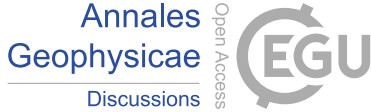

An example case gave an unfounded change of 2.45 mV/m (magnetic storm level according to
Troshichev et al., 2017) at local midnight and hardly any effect at noon at 4 nT amplitude in
smoothed IMF $B_Y$ values which is a common occurrence (Stauning, 2015).


**53  2. Calculation of IMF BY-related solar wind sector term**

The commented publication, J&T2011, holds (p.1496-97) a step-by-step procedure quoted below
for near-real time calculations of IMF $B_Y$-related solar wind sector (SS) terms by forward cubic
spline-based extrapolation of past median values:
*"Keeping in mind this specification, the 3-day smoothing averages of the median values were subjected to*
*the interpolation procedure including the following steps:*
*1. median values for magnetic components H and D are derived for 4 intervals of days preceding with the*
*exception of the current day (n=0):*
*-      r1=F[for interval from n-3 day to n-1 day]*
*-      r2=F[for interval from n-5 day to n-3 day]*
*-      r3=F[for interval from n-7 day to n-5 day]*
*-      r4=F[for interval from n-9 day to n-7 day];*
*2. piecewise polynomial form of the cubic spline interpolant for r1, r2, r3, and r4 segments is determined;*
*3. termination of this form related to day n=0 is examined as representative of the SS effect for the current*
*day, even if this day is disturbed.*
*The procedure is repeated each subsequent day. Results of the procedure – the variation of the reconstructed magnetic*
*H component is presented by the magenta line in the same Fig. 6, the reconstructed H-component curve being shifted by*
*50 nT to a lower position"*
Thus, it is stated (p. 1497) that this procedure was used to derive the smoothly varying display of
the H-component SS-term (magenta line) in their Fig. 6 using magnetic data from Qaanaaq (THL)
for the interval 1-30 June 2001, here reproduced in Fig. 1 and re-calculated in Fig. 2. However, the
statement concluding the quoted procedure is incorrect. The SS-term ($H_{SS}$) displayed in Fig. 6 of
J&T2011 could not have been generated by the quoted near-real time procedure. The smoothed
magenta curve for $H_{SS}$ is not a real-time version but derived by using the post-event method based
on daily median values smoothed over 7 days with the actual day at the middle. Fig. 2 presents the
post-event ("final") $H_{SS}$ values derived by the PCN index suppliers at the Danish Space Research
Institute (DTU Space).
Values of the solar wind sector term, $H_{SS}$, derived by adhering rigorously to the above quoted
"near-real time" procedure (including the cubic spline-based forward projection) are displayed by
the jagged curve in magenta line in Fig. 3.



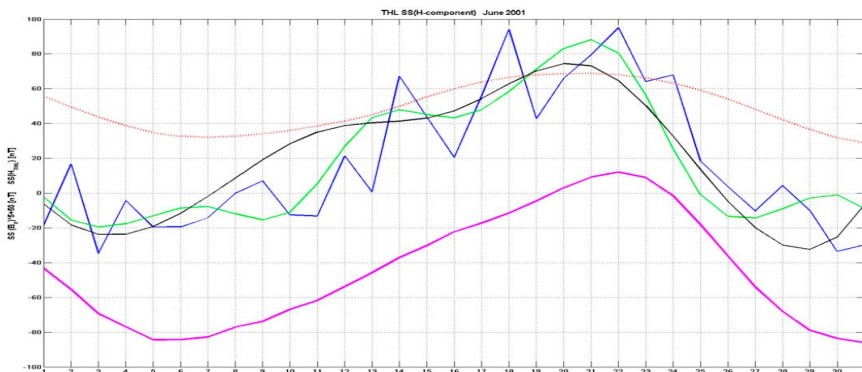


**Fig. 6.** Behavior of the median values of the magnetic H-component at Thule station during June months of 1998 **(a)** and 2001 **(b)** for intervals with duration of 1 day (blue line), 3 days (green line), and 5 days (black line). The red dotted line shows the variation of the IMF $B_y$ component, derived from spacecraft measurements. The magenta line shows the variation of the reconstructed magnetic H-component. To be clearly demonstrated, the actual $B_y$ values were multiplied by five and were shifted by 50 nT to a higher position, whereas the curve of reconstructed H-component was shifted by 50 nT to a lower position.

**Fig. 1**. THL H-component. 1-day (blue line), 3-days (green) and 5-days (black) median values. Resulting $H_{SS}$
terms in magenta line on a scale shifted 50 nT downward for clarity. Smoothed IMF By multiplied by 5 and
shifted 50 nT upward in red line. (Reproduced from Fig. 6b of Janzhura and Troshichev, 2011, including
caption).

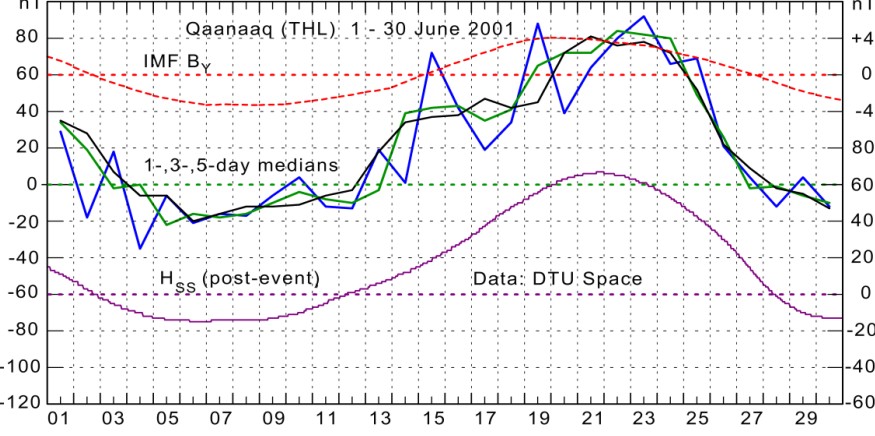

**Fig. 2.** THL 1-, 3-, 5-day medians on left scale. Post-event $H_{SS}$ terms are displayed in magenta line on lower
right scale. The data were supplied from DTU Space. Smoothed IMF $B_Y$ values (red line) added on upper
right scale (data from OMNIweb).



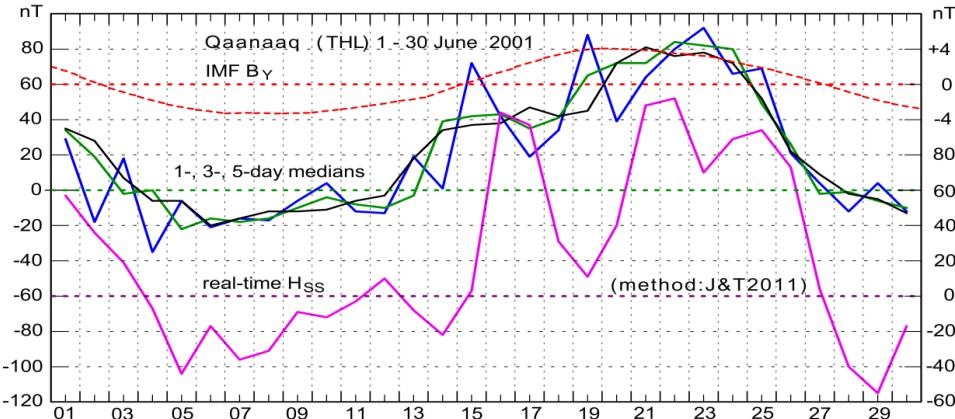

**Fig. 3.** THL 1-, 3-, 5-day medians on left scale. Simulated real-time $H_{SS}$ terms in magenta line on lower
right scale. $H_{SS}$ values were calculated by following exactly the procedure in J&T2011.(Stauning, 2018a).
Smoothed IMF $B_Y$ values added on upper right scale (OMNIweb)

The similarity between the $H_{SS}$ curves in Figs.1 and 2 and the large difference with respect to the
simulated real-time $H_{SS}$ values in Fig.3 derived from cubic spline-based extrapolation of past
median values implies that the display in Fig. 6 of J&T2011 (Fig. 4.15 of T&J2012), contrary to the
statement in p. 1496-97, was actually generated by using post-event calculations with smoothed
averages of 7 days daily median values.

**3. Use of Solar Wind Sector (SS) term in reference level for PC indices.**
The IMF By-related solar wind sector effects on the convection patterns generate changes in the PC
index response to the merging electric fields. The solar wind sector (SS) term was implemented in
the derivation of PC index reference levels by J&T2011. The SS-term from their Fig. 6 has been
added to the quiet day variation (QDC) with slowly (seasonally) varying amplitude calculated by
the method published in Janzhura and Troshichev (2008) to generate the June section (days 152-
182) of the reference level displayed by the solid line superimposed on the 1-min H-component
values displayed in their Fig.1. The remaining part of the reference levels has no doubt been
generated by the same method reproduced here in Fig. 4 (including caption) and recalculated in Fig.
5. However, using the near-real time $H_{SS}$ version generates the jagged reference level displayed in
red solid line superimposed on the H-component values displayed in faint blue line in Fig. 6.



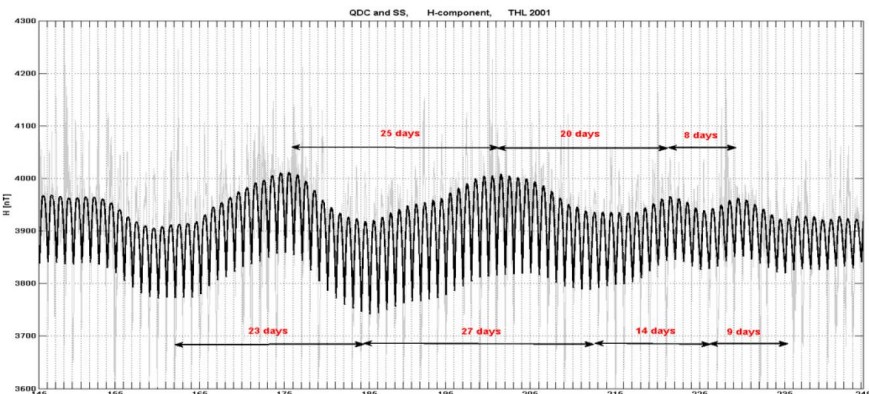


**Fig. 1.** Superposition of the actual variation of 1-min values of the geomagnetic H-component observed at Thule station in the summer season of 2001 (thin lines) and the quiet daily curve (QDC) characterizing the daily variation of the quiet geomagnetic field (thick solid lines).


**Fig. 4**. PCN reference level (thick line) superimposed on recorded H-component data (thin line).
Reproduction of Fig. 1 of Janzhura and Troshichev (2011), (caption included).

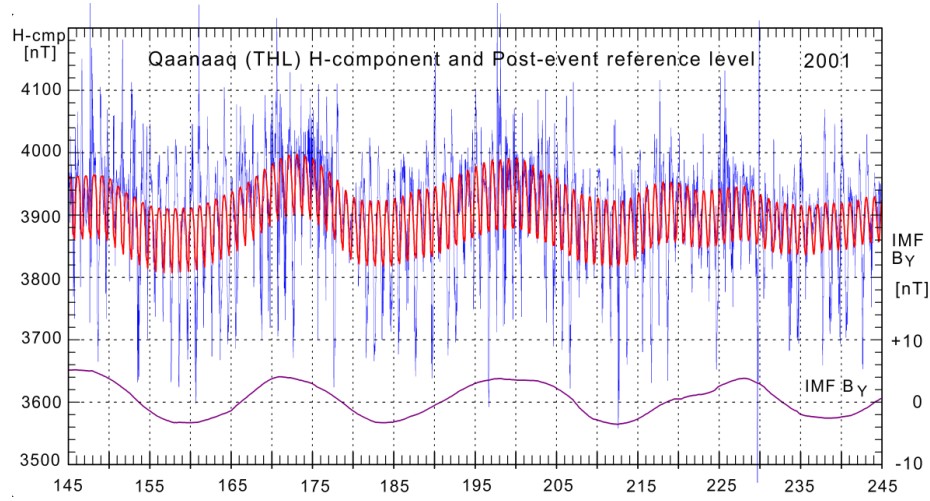


**Fig. 5**. Post-event (final) PCN reference level values (red line) supplied from DTU Space
superimposed on recorded H-component data (blue line). IMF By values (magenta line) on the
lower right scale added at the bottom of the diagram for reference. (after Stauning, 2015).





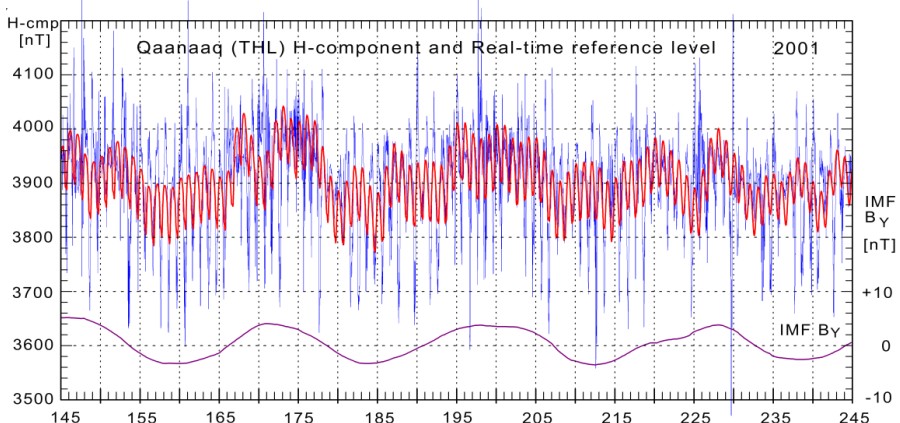

**Fig. 6**. Simulated real-time PCN reference level (red line) derived by rigorous use of the J&T2011 procedure superimposed on recorded H-component data (blue line). IMF By values added at the bottom of the diagram.

The close similarity of the reference level (thick line) in Fig. 4 with that of Fig. 5 and the strong difference with respect to the real-time reference level in Fig. 6 implies that the "QDC" in Fig. 4 was actually derived by the post-event (final) calculation method (7-days smoothing of daily median H-component values) like the method used for Fig. 5.

**4. The SS-effects throughout a year (2001).**

Figure 7b in J&T2011 (Fig. 4.16b in T&J2012) displays the H-component recorded at Qaanaaq (THL) throughout year 2001. The IMF $B_Y$-related $H_{SS}$ values have been superimposed on the recordings. According to the description, the black asterisks present near-real time values, while the red asterisks present post-event values. It appears that the two sets of symbols merge to form a smooth continuous curve. However, contrary to the description in text and figure caption, both symbol series have been generated by post-event calculation methods (averaged daily median values).

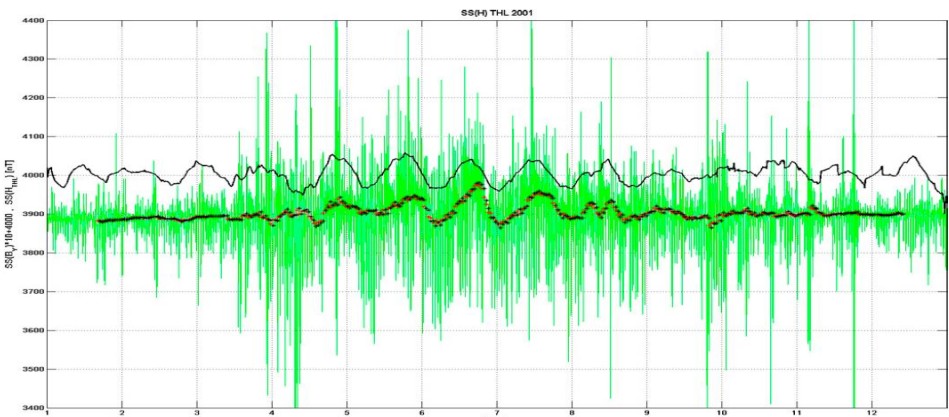

Fig. 7. The SS effect derived in the H-component observed at station Thule in 1998 (a) and 2001 (b). The actual variation of the ground H-component is shown by the green line, whereas black asterisks present the extrapolated SS structure obtained by the extrapolation procedure when all data are available till the examined day ($n = 0$), and red asterisks present the interpolated SS structure derived under the condition that the examined day is in the middle of a gap in the time interval. The actual variation of the IMF $B_y$ component, measured by ACE spacecraft, is shown by the thin black line.

**Fig. 7.** Presentation of one year's THL H-component data (green line) with superimposed near-real time
(black), and post-event $H_{SS}$ values (red asterisks). IMF By values added (black line). Reproduced from Fig.
7b of Janzhura and Troshichev (2011) (caption included).
Re-calculated values are displayed in Fig. 8 where the post-event symbols (magenta diamonds)
merge to form a continuous broad trace of $H_{SS}$ values. The scattered near-real time symbols (black
diamonds) have been calculated by using rigorously the J&T2011 near-real time procedure quoted
above.

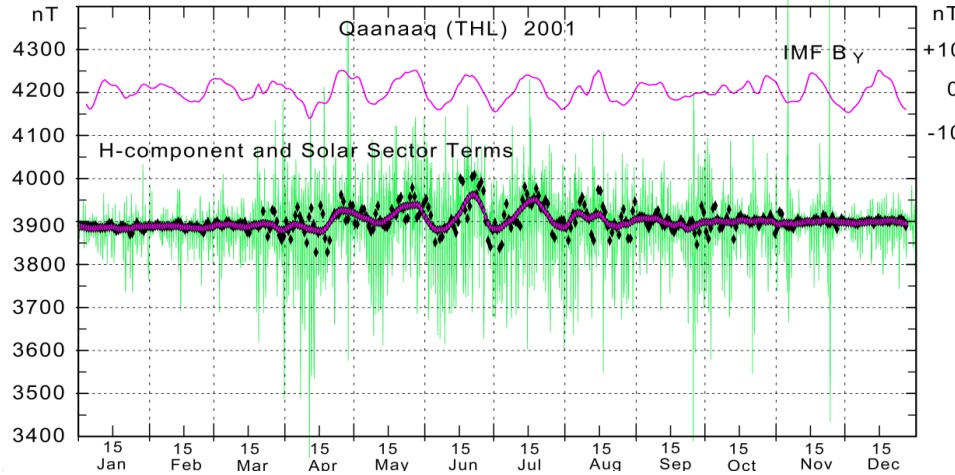

**Fig. 8.** THL H-component data with superimposed  simulated real-time (black), and post-event (magenta
diamonds) values of $H_{SS}$. Note the large scatter of the real-time (black) diamonds. IMF By values added (red
line) on upper right scale.






Figs. 9a, b provide more detailed comparisons of the values displayed in Figs. 7 and 8.

**a.**                                              **b.**

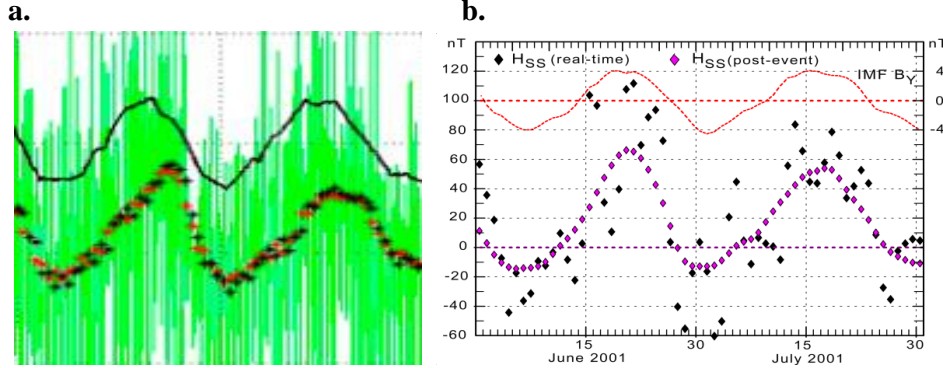

**Fig. 9.** (a) Detailed plot of June-July section of Fig. 7 from Janzhura and Troshichev (2011). Note the almost
continuous transition from black to red diamonds. IMF By values added (black line). (b) Details of Fig. 8.
Note the large scatter in the black diamonds (simulated real-time) away from the post-event magenta
diamond symbols. IMF By values added for reference (red line).

It is evident from Figs. 8 and 9b that the real-time and post-event methods generate quite different
values of the IMF $B_Y$-related solar wind sector terms, $H_{SS}$. The set of red asterisks in Figs. 7 and 9a,
no doubt, present post-event (smoothed daily median) values, while the black set in Figs. 7 and 9a,
against the statements in the text and in the figure caption, could not present near-real time values
but must have been derived by post event smoothing.

**5. Identification of solar wind sector structure in near real time**
Fig. 8b of J&T2011 reproduced here in Fig .10 display close relations between solar sector terms
$D_{SS}$ and $H_{SS}$ and daily average IMF $B_Y$ values. These relations are assumed to enable the possible
identification of the IMF sector structure in near-real time (title of the publication).

**a.**                                              **b.**

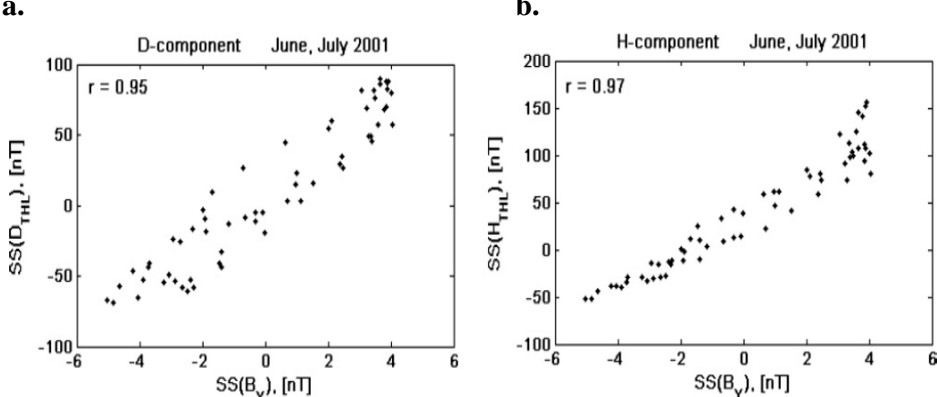

**Fig. 8.** The relationship between the satellite and ground-based sets of magnetic data on variations caused by the IMF sector structure in the
        summer months of 1998 (upper row) and 1991 (lower row) for geomagnetic D (left column) and H (right column) components at Thule.


**Fig. 10**. Reproduction of Fig. 8b from Janzhura and Troshichev (2011) (incl. caption). Relations between daily average IMF $B_Y$ values and solar wind sector (a) $D_{SS}$ and (b) $H_{SS}$ values. Note: Scale values are by some error (misprint) too large by a factor 2.


$D_{SS}$ and $H_{SS}$ values for the summer months, June-July, 2001, provided by DTU-Space are displayed in Figs. 11a,b (left and right). The values have been calculated from the 7-days averaged daily median D- and H-component values using the post-event method.


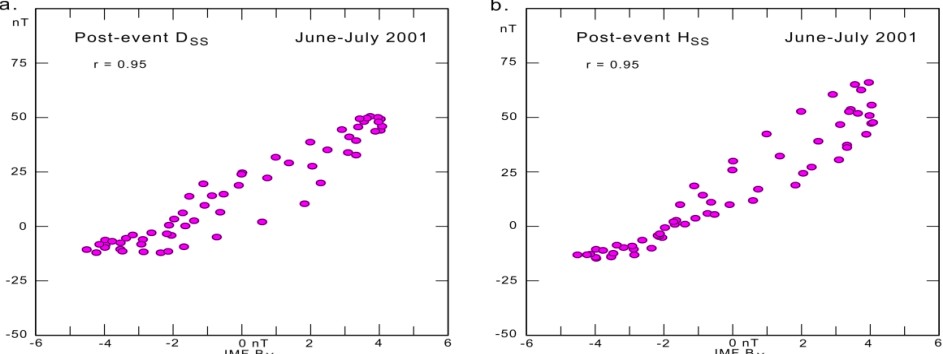


**Fig. 11**. Display of post-event (final) solar wind sector terms, $D_{SS}$, $H_{SS}$, supplied from DTU Space vs. daily average IMF $B_Y$ values. Note, that the scales are reduced by factor 2 from those of Fig. 10 to correct the scaling errors in Fig. 8 of J&T2011.

The corresponding near-real time values of $D_{SS}$ and $H_{SS}$ for June-July, 2001, have been calculated by rigorous use of the above quoted near-real time procedure from J&T2011. The results are displayed in Figs. 12a,b.

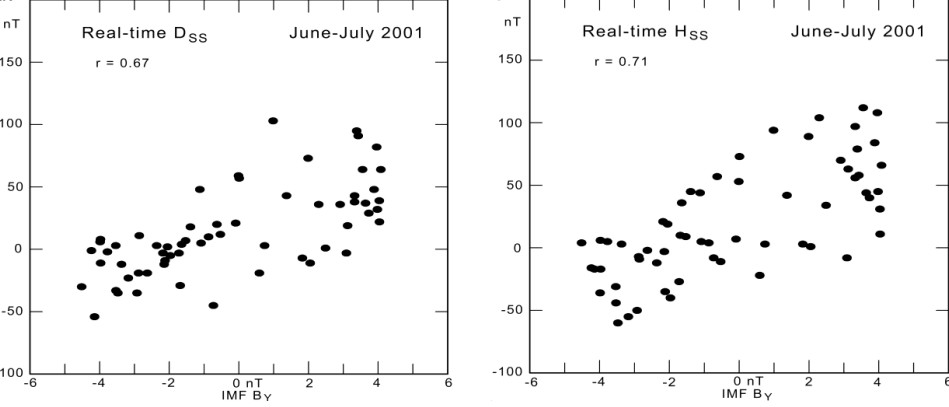

212
213

**Fig. 12**. Display of simulated real-time solar wind sector terms, $D_{SS}$, $H_{SS}$, calculated by using the procedure in J&T2011, p.1496, plotted vs. daily average IMF $B_Y$ values.

The similarity between the diagrams of Fig. 11 definitely constructed by post-event calculations (at DTU Space) and those of Fig. 10 indicates beyond doubt that the latter have been derived by post-




event methods. From the post-event processing displayed in Figs. 10 and 11, the D- and H-
component solar wind sector terms appear highly correlated (r = 0.95, 0.97) with the daily mean
IMF $B_Y$ values. Thus, they could be used to estimate past IMF $B_Y$ levels and signs with good
probability from archived data. However, the objective according to the title and abstract of the
paper was to estimate IMF $B_Y$ in near real time.
Compared to the $D_{SS}$ and $H_{SS}$ solar wind sector terms derived by post-event calculations displayed
in Figs. 10 and 11, the corresponding solar wind sector terms generated by using real-time
processing are much less well correlated with the daily average IMF $B_Y$ values (r = 0.67, 0.71). The
relations displayed in Fig. 12 by the scattered $D_{SS}$ and $H_{SS}$ values derived by using near-real time
methods could hardly be used to determine the actual IMF $B_Y$ magnitude level and sign with any
certainty, which was the main scope of the J&T2011 publication.

**Summary and conclusion**
The commented paper, J&T2011, and its replica in Troshichev and Janzhura (2012), are significant
since along with the publications Troshichev et al. (2006) and Troshichev et al. (2011) held in
chapter 4 of Troshichev and Janzhura (2012), they form the basis for the derivation procedures
(Matzka, 2014; Nielsen and Willer, 2019) applied for calculations of Polar Cap (PC) index values in
the near-real time and post-event (final) versions endorsed by IAGA Resolution #3 (2013).
However, neither the illustrations nor the results presented in J&T2011 have been derived by the
specified near-real time method. All the illustrations and results presented in Figs. 1, 6, 7, and 8
display values derived by post-event calculation methods based on daily median values smoothed
over 7 days centred on the day of interest. Figs. 2, 3, and 4 display observed values smoothed over 7
days, while the remaining Fig. 5 displays averages over 4 months.
In summary, there is strong disagreement between indications in the title, abstract, statements in the
text, and captions, as well as in the presentation of results, compared to re-calculations by rigorous
use of the presented near-real time procedure. Thus, it is concluded to caution against uncritical use
of the methods and results presented in the commented publication by Janzhura and Troshichev
247 (2011).

**Conflicts of interests**. The author declares that he has no conflict of interest with respect to the
present work.

**Data availability**
Geomagnetic data from Qaanaaq (THL) were supplied from the INTERMAGNET data service web
portal at http://intermagnet.org.
Solar wind plasma and magnetic field data based on data from the ACE, IMP, GeoTail, and WIND
space missions were supplied from the OMNIweb data service at http://omniweb.gsfc.nasa.gov .
Interim values of solar wind sector, $H_{SS}$ and $D_{SS}$, and quiet day, QDC, values from PCN
calculations for 2001 were supplied by the index providers at DTU Space.





**Acknowledgments.** The staffs at the observatory in Qaanaaq its supporting institutes, the Danish
Meteorological Institute (DMI) and the Danish Space Research Institute (DTU Space), are
gratefully acknowledged for providing high-quality geomagnetic data for this study. The efficient
provision of geomagnetic data from the INTERMAGNET data service centre, the supply of solar
wind data from the IM8, WIND, GeoTail, and ACE missions, and the excellent performance of the
OMNIweb data portals are greatly appreciated. The author gratefully acknowledges the
collaboration and many rewarding discussions in the past with Drs. O. A. Troshichev and A. S.
Janzhura at the Arctic and Antarctic Research Institute in St. Petersburg, Russia.

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
