# Peer review of "Title: Comment on "Identification of the IMF sector structure in near-real time by ground magnetic"

_Annales Geophysicae, 2020_

## Short Comment (SC1) · 15 Jan 2021

Comments to MS https://doi.org/10.5194/angeo-2020-53 Title: Comment on "Identification of the IMF sector structure in near-real time 5 by ground magnetic data" by Janzhura and Troshichev (2011). Author: Peter Stauning, Danish Meteorological Institute, Lyngbyvej 100, Copenhagen.

MS is devoted to analysis of discrepancies in results of identification of the IMF sector structure by methods put forward by Janzhura and Troshichev [2011] and by Stauning

[2011]. Author (Dr. P.Stauning) (a) expresses the criticism concerning the method and results presented in [Janzhura and Troshichev, 2011], (b) identifies this method with the unified PC derivation method [Troshichev et al., 2006], and (c) makes the conclusions that "the IAGA endorsed PC index is to a large extent based on the methods described in Janzhura and Troshichev (2011)". In this connection it should be noted that (1) method of the IMF SS identification [Janzhura and Troshichev, 2011] has no any relation to the unified PC derivation procedure [Troshichev et al, 2006]. Dr. Stauning could not understand this principal fact during elapsed 10 years suggesting his scientific insolvency, (2) the unified PC derivation method [Troshichev et al, 2006] was approved by IAGA Division V-DAT at special meeting in Vienna in May 2010. Just this Division V-DAT recommendation was a basis for the IAGA endorsement. This fact is well known to Dr.Stauning and suggests his scientific unscrupulousness. I recommend to decline the manuscript of Dr.Stauning taking into account that he has published the papers with the same content [Stauning, 2013b, 2015, 2018a,c, 2020].

---

## Author Comment (AC1) · 16 Jan 2021

Copenhagen 16 January 2021/PSt

Reply to Interactive comments by Dr. O.A. Troshichev on "Comment on "Identification of the IMF sector structure in near-real time by ground magnetic data" by Janzhura and Troshichev (2011)" by Peter Stauning"

The interactive comments from Dr. Troshichev mentions: (a) the criticism concerning the method and results presented in [Janzhura and Troshichev, 2011] Reply: Yes, The

abstract and the conclusions in my commentary state that "neither the illustrations nor the results presented there have been derived by the specified near-real time method. Figs. 1, 6, 7, and 8 display values derived by post-event calculations based on daily medians smoothed over 7 days centred on the day of interest." I see no objection against this statement in the present comment from Dr. Troshichev.

(b) identifies this method with the unified PC derivation method [Troshichev et al., 2006] Reply: No. There is no such statement in my commentary.

(c) makes the conclusion that the IAGA endorsed PC index is to a large extent based on the methods described in Janzhura and Troshichev [2011]. Reply: Yes.The abstract and conclusion in my commentary state that "The commented paper, J&T2011, and its replica in Troshichev and Janzhura (2012), are significant since along with the publications Troshichev et al. (2006) and Troshichev et al. (2011) held in chapter 4 of Troshichev and Janzhura (2012), they form the basis for the derivation procedures (Matzka, 2014; Nielsen and Willer, 2019) applied for calculations of Polar Cap (PC) index values in the near-real time and post-event (final) versions endorsed by IAGA Resolution #3 (2013)." IAGA Resolution #3 (2013) was agreed at the IAGA Assembly in 2013 in response to the application for endorsement submitted jointly from AARI and DTU Space on 25 February 2013 and recommended by the Task Force comprising Drs. Menvielle, McCready, and Demetrescu. Their statement from 20 August 2013 reads: "The PC index being recommended for endorsement at IAGA 2013 in Merida, Mexico is that defined by the following publications: Troshichev et al. (2006 and 2009), Janzhura and Troshichev (2008), Janzhura and Troshichev (2011)". The material presented to IAGA for the endorsement and description of the methods actually used for calculations of the "definitive" PCN indices are available at the DTU Space web portal: ftp://ftp.space.dtu.dk/WDC/indices/pcn I shall call on representatives from DTU Space, Drs. N. Olsen and A. Willer, to provide verification of the above statements.

Copenhagen 16 January 2021

[Figure]

Peter Stauning

---

## Author Comment (AC2) · 30 Jan 2021

Copenhagen 30 January 2021/PSt

Final Author Comments to ANGEOD discussions on angeo-2020-53

(1) Summary of submitted contribution: P. Stauning: "Comment on "Identification of the IMF sector structure in near-real time by ground magnetic data" by Janzhura and Troshichev (2011)." Angeo-2020-53.

[Figure]

The main issue in the commentary is the ascertainment that the illustrations and results presented in Janzhura and Troshichev (2011) (J&T2011) are based on post-event methods in spite of the stated aims in the title, the abstract and in the descriptions of methods to calculate the effects of the solar wind sector structure on polar cap magnetic variations in near-real time.

The contributions from the solar wind sector structure (SS) are important for the calculations of the reference level ("QDC") used in the derivation of the polar magnetic variations providing the basis for polar cap (PC) index values whether in real time or post-event applications.

The post-event QDC methods are used for the "definitive" PC index calculations made at the Arctic and Antarctic Research Institute (AARI) and the Danish Space Research Institute (DTU Space). The near-real time QDC calculations are important for the use of PC indices in space weather monitoring applications. Thus, the J&T2011 publication is unfortunate for developments of true real-time methods as well as the consolidation of definitive PC index calculations.

As stated in the conclusions: "The commented paper, J&T2011, and its replica in Troshichev and Janzhura (2012), are significant since along with the publications by Troshichev et al. (2006) and Troshichev et al. (2011) held in chapter 4 of Troshichev and Janzhura (2012), they form the basis for the derivation procedures (Matzka, 2014; Nielsen and Willer, 2019) applied for calculations of Polar Cap (PC) index values in the near-real time and post-event (definitive) versions endorsed by IAGA Resolution #3 (2013).

However, neither the illustrations nor the results presented in J&T2011 have been derived by using the specified near-real time methods. The illustrations and results presented in Figs. 1, 6, 7, and 8 display values generated by post-event calculation methods based on using solar sector terms derived from daily median values smoothed over 7 days centred on the day of interest. Figs. 2, 3, and 4 display observed values

smoothed over 7 days, while the remaining Fig. 5 displays averages over 4 months".

(2) Summary of Interactive Comments angeo-2020-53-SC1.pdf & Reply angeo-2020-53-AC1.

In his interactive comment, angeo-2020-53-SC1.pdf, Dr. Troshichev, corresponding author of the commented publication, Janzhura and Troshichev (2011), and first author of its replica in ch. 4.4 "Allowance for IMF sector structure" of Troshichev and Janzhura (2012), makes no attempt to contradict the statements of mingled QDC methods. On the contrary, it seems that Dr. Troshichev denounce the importance of the commented publication (J&T2011) for the IAGA-endorsed PC index calculations whereby the "definitive" PCN index series extending from 1975 to present calculated by DTU Space is declared invalid.

Further in his interactive comment, angeo-2020-53-SC1.pdf, Dr. Troshichev argues that the method described in Troshichev et al. (2006) "was approved by the IAGA Division V-DAT at a special meeting in Vienna in May 2010". This argument is not seen to be relevant for the present discussion since the Troshichev et al. (2006) methods are not mentioned at all in my submission. The statement, furthermore, is incorrect. According to IAGA V-DAT there was no formal (business) V-DAT meeting in 2010 as such meetings are held during IAGA Assemblies, e.g., in 2009 or 2011. Furthermore, there is no IAGA documentation from a V-DAT meeting held in May 2010.

In his interactive comment, Dr. Troshichev repels publishing the submitted commentary and avoids offering corrections of the commented publication by Janzhura and Troshichev (2011). The bottom line is still that against the title, abstract, and specific statements on the use of near-real time methods, the illustrations and results presented in Figs. 1, 6, 7, and 8 display values derived by post-event QDC calculation methods based on daily median values smoothed over 7 days centred on the day of interest which excludes real-time application.

In summary, it is concluded that the submitted contribution, angeo-2020-53, should be

published to rectify the inadequate illustrations and to caution against uncritical use of the methods and results presented in the commented publication by Janzhura and Troshichev (2011).

Copenhagen 30 January 2021

Peter Stauning pst@dmi.dk

References not included in the submitted commentary.

Nielsen, J. B. and Willer, A. N.: Restructuring and harmonizing the code used to calculate the Definitive Polar Cap Index, Report from DTU Space, 2019. Available at: ftp://ftp.space.dtu.dk/WDC/indices/pcn/ .

Troshichev, O. A., Podorozhkina, N. A., and Janzhura, A. S.: Invariability of relationship between the polar cap magnetic activity and geoeffective interplanetary electric field, Ann. Geophys., 29, 1479-1489, https://doi.org/10.5194/angeo-29-1479-2011, 2011.

---

## Author Comment (AC3) · 5 Feb 2021

Copenhagen 5 February 2021/PSt

Documentation on Janzhura and Troshichev (2011) and relations to Polar Cap (PC) indices.

1. Introduction.

The publication Janzhura and Troshichev (2011): "Identification of the IMF sector structure in near-real time by ground magnetic data" is important because it constitute an essential reference for the methods used for derivation of Polar Cap (PC) indices in real-time as well as in definitive versions. The very particular feature of this publication is the dual approach that the specified method for calculation of the so-called solar wind sector terms Hss and Dss advocates real-time methods while the illustrations display values derived by post-event methods. The solar wind sector terms provide essential parts of the reference level used for definition of the magnetic disturbances which are basis for the PC indices.

2. The dual approach in Janzhura and Troshichev (2011)

The real-time approach to derive the solar wind sector terms uses cubic spline-based forward extrapolation based on daily median component values throughout 9 days before the actual day while the post-event method use simple 7-days "box averaging" of daily median values with the day in question at the middle. The latter method is clearly not suited for real-time applications but provides nice illustrations while the real-time method sounds complicated enough to deter from re-calculations that would give bad-looking illustrations.

This dual approach has confused IAGA officials as well as editors and reviewers over the years. In addition, the acceptance by IAGA of the Janzhura and Troshichev (2011) publication as reference for the PC index endorsement has blocked for a thorough examination of methods and development of improved calculation procedures. The dual approach is illustrated in the definition of method in pp. 1496-1497 and the illustration of results in Fig. 6 of Janzhura and Troshichev (2011) summarized in Fig. 1 here.

Fig. 1. Essential features of Janzhura and Troshichev (2011): "Identification of the IMF sector structure in near-real time by ground magnetic data".

Figure 1 here displays a summary of essential features of Janzhura and Troshochev (2011) such as the abstract and the specification of the near-real time procedure for deriving the solar wind sector (Hss) term using cubic spline-based forward extrapolation

of median H-component values from the previous 9 days. According to the statements in the text (p. 1496-1497) this procedure was used to generate the display in their Fig. 6. Actually, recalculations indicate that the Hss term in the display was generated by simple "box-averaging" of 7 daily median values at a time around the day in question (see Fig. 2)

3. Stauning (2013).

The publishing of Janzhura and Troshichev (2011) ahead of the IAGA Assembly in Mexico 2013 spurred a critical comment (Stauning, 2013) submitted to Annales Geophysicae in November 2012, published July 2013 and circulated to IAGA officials, among others to the IAGA Task Force members, prior to the IAGA Assembly. Citations from Stauning (2013): "Comments on quiet daily variation derivation in "Identification of the IMF sector structure in near-real time by ground magnetic data" by Janzhura and Troshichev (2011)":

Abstract. Comments on the QDC derivation described in: Janzhura, A. S., Troshichev, O. A. (2011): Identification of the IMF sector structure in near-real time by ground magnetic data, Ann. Geophys., 29, 1491-1500, doi:10.5194/angeo-29-1491-2011. The description presented in the paper of the relations of the solar wind sector structure to the derivation of the quiet daily variation (QDC) in polar magnetic recordings used for calculation of Polar Cap (PC) indices is found to be unclear and not properly justified. The presented example on inclusion of a solar sector term in an actual QDC series is found to be questionable even on the authors' premises.

Conclusions. The new definition includes a solar wind sector (SS) IMF BY-related term. The inclusion of this term is not adequately described and justified and the resulting inclusion of a SS term in the QDC level is inconsistent even on the authors' own premises. The resulting QDCs for the H-component (their Figure 1) display a strong SS IMF BY-related modulation in the level defined during local night in spite of the evidence presented (their Figure 5) that the nighttime polar magnetic H-component

values are not influenced much by IMF BY variations.

Acknowledgements: Topical Editor R. Nakamura thanks B. Emery for her help in evaluating this paper.

It should be mentioned that the topical editor requested comments from Dr. Troshichev. The request was not answered.

4. Application for PC index Endorsement by IAGA submitted February 2013.

Citations from Matzka (2014):

"This text is based on the "Relevant supporting material " as sent to IAGA on 25/02-2013 and describes the IAGAendorsed PC index. It is prepared by Dr. Oleg Troshichev, Dr. Alexander Janzhura and Dr. Jürgen Matzka.

Regarding criterion 2:

The derivation of the index is described in the following publications: Troshichev et al. (2006) Janzhura and Troshichev (2008) Janzhura and Troshichev (2011) Troshichev and Janzhura (2012) (here, chapter 4 describes derivation of the provisional data set)"

5. Task Force recommendation (20 Aug. 2013).

Citations of text of recommendation:

Recommendation by the Task Force: Fully recommend endorsement of the PC index.

Members of the taskforce: Michel Menvielle, Heather McCreadie and Crisan Demetrescu. ("We" in this document refers to the task force)

The PC index being recommended for endorsement at IAGA 2013 Merida, Mexico is that defined by the following publications: Troshichev et al. (2006 and 2009), Janzhura and Troshichev (2008), Janzhura and Troshichev (2011)".

6. IAGA Resolution #3 (2013)

A IAGA Business Meeting during the 2013 Assembly adopted the Task Force recommendation and agreed on the proposed text for Resolution #3 (2013). The resolution was later passes by IAGA Executive Committee headed by Professor Mioara Mandea.

7. Stauning (2015).

The publication by Janzhura and Troshichev (2011) has formed the PC index calculation procedures, among others those implemented in the PCN calculations at DTU Space.

Further critical notes have been published such as: Stauning (2015): "A Critical Note on the IAGA-endorsed Polar Cap Index Procedure: Effects of solar wind sector structure and reverse polar convection". This publication performed a quantitative assessment of the consequences of using the post-event methods. It was mentioned in the abstract and conclusions that "The added IMF By-related terms may introduce unjustified contributions to the PC index of more than 2 index units (mV/m)".

The topical editor, Dr. G. Balasis, invited without success the authors of Janzhura and Troshichev (2011) to submit their views and comments.

8. Stauning (2018).

The effects of using the devised near-real time method defined in Janzhura and Troshichev (2011) was assessed in the publication: P. Stauning (2018): "A critical note on the IAGA-endorsed Polar Cap (PC) indices: excessive excursions in the real-time index values".

It is mentioned in the abstract that "The present note provides the first reported examination of the validity of the IAGA-endorsed method to generate real-time PC index values. It is demonstrated that features of the derivation procedure defined by A. S. Janzhura and O. A. Troshichev in Ann. Geophys, 29, 1491-1500 (2011) may cause considerable excursions in the real-time PC index values compared to the final index values. In examples based on occasional downloads of index values, the differences

between real-time and final values of PC indices were found to exceed 3 mV/m, which is a magnitude level that may indicate (or hide) strong magnetic storm activity."

The definition in Janzhura and Troshichev (2011) of the real-time method is again quoted:

Keeping in mind this specification, the 3-day smoothing averages of the median values were subjected to the interpolation procedure including the following steps:

1. median values for magnetic components H and D are derived for 4 intervals of days preceding with the exception of the current day (n=0):

- r1=F[for interval from n-3 day to n-1 day]

- r2=F[for interval from n-5 day to n-3 day]

- r3=F[for interval from n-7 day to n-5 day]

- r4=F[for interval from n-9 day to n-7 day];

2. piecewise polynomial form of the cubic spline interpolant for r1, r2, r3, and r4 segments is determined;

3. termination of this form related to day n=0 is examined as representative of the SS effect for the current day, even if this day is disturbed.

The procedure is repeated each subsequent day. Results of the procedure – the variation of the reconstructed magnetic H component is presented by the magenta line in the same Fig. 6, the reconstructed H-component curve being shifted by 50 nT to a lower position.

Here, Fig. 2 illustrates the different results obtained by using the prescribed real time method illustrated by the broken dashed line and the results displayed in Fig. 6 of Janzhura and Troshichev (2011) here shown by the full smooth curve in magenta line.

Figure 2. The 3-days median values (from Fig. 6b of J&T2011) are shown in green line.

The Hss values from Fig. 6b of J&T2011 are shown by the smooth heavy magenta line on the scale to the right, while the Hss values calculated here by Cubic Spline extrapolation are shown on the same scale by dots connected by the dashed magenta line. (copy of Fig. 4 of Stauning, 2018)

Supplementary data files (references and examples):

IAGA PC_index_description_main_document.pdf (12-02-2014)

IAGA PC_index_description_appendix_A.pdf (27-01-2014)

IAGA PCS Oct-Nov 2014 prompt data: pcnpcs2014.zip (download 11-11-2014 09:41)

IAGA PCS Oct-Nov 2014 final data: pcnpcs2014.zip (download 25-10-2017 11:32)

DMI PCS Oct-Nov 2014 prompt data: PCS14C.5QP

DMI PCS Oct-Nov 2014 final data: PCSU2014.5MQ

These files are available at: https://doi.org/10.5194/angeo-36-621-2018-supplement .

Topical editor, Dr. Anna Milillo, attempted without success to obtain comments from Dr. Troshichev, corresponding author of Janzhura and Troshichev (2011).

9. IAGA response

The concerns over the adverse results from using the real-time (cubic spline-based extrapolation of previous daily medians) were forwarded to IAGA. In response, a letter from Secretary General, professor Mioara Mandea was received. The essential points of the letter read:

"Thank you for your recent email in which you raised objections to the current method of deriving the PC index. Having now had the opportunity to have a thorough read of your documents I believe that you are not making a new objection, but rather are restating earlier objections, which you have raised with several people associated with IAGA over several years (and which we discussed a lot about in 2014).

The IAGA Executive Committee and Division Leaders have discussed this issue and your comments and we have concluded that this subject can only be reopened for scrutiny if something is published in scientific literature to which IAGA would have to respond. Otherwise, a unanimous vote to endorse the PC index was held in 2013 and we therefore consider that no further discussions or reviews are required at this time. In the meantime, where the index is listed only as provisional or quick-look, then users of the index should be aware of the risk of using it and not rely on a provisional or quick-look index for definitive science."

Noting in a comment to the above statements that IAGA Resolution #3 (2013) recommends the use of the near-real time as well as the definitive versions. Furthermore, the IAGA-endorsed PCN version calculated at DTU Space is declared "definitive".

10. Stauning (2020)

A recent further comment to Janzhura and Troshichev (2011) by P. Stauning (2020): "The Polar Cap (PC) index: invalid index series and a different approach". Article DOI: 10.1029/2020SW002442 examines the real-time as well as the post-event methods.

It is concluded that the use of the post-event method implied by Fig. 6 of Janzhura and Troshichev (2011) may generate errors in the derived PCN index values by more than 3 mV/m (magnetic storm level) while the use of the real-time method defined by the instructions in Janzhura and Troshichev (2011) may generate additional excessive excursions of up to 3 mV/m.

The topical editor, Dr. Mike Hapgood, tried to obtain comments from Dr. Troshichev offering him space to present his views and delayed the publication of the article by several months to await his reply which, unfortunately, never arrived.

References:

Janzhura, A. S., Troshichev, O. A. (2011). Identification of the IMF sector structure in near-real time by ground magnetic data. Annales Geophysicae, 29, 1491-1500.

https://doi.org/10.5194/angeo-29-1491-2011 .

Matzka, J. (2014). PC_index_description_main_document_incl_Appendix_A.pdf. Available at DTU Space web portal: ftp://ftp.space.dtu.dk/WDC/indices/pcn/

Stauning, P. (2013). Comments on quiet daily variation derivation in "Identification of the IMF sector structure in near-real time by ground magnetic data" by Janzhura and Troshichev (2011). Annales Geophysicae, 31, 1221-1225. https://doi.org/10.5194/angeo-31-1221-2013 .

Stauning, P. (2015). A critical note on the IAGA-endorsed Polar Cap index procedure: effects of solar wind sector structure and reverse polar convection. Annales Geophysicae, 33, 1443-1455. https://doi.org/10.5194/angeo-33-1443-2015 .

Stauning, P. (2018). A critical note on the IAGA-endorsed Polar Cap (PC) indices: excessive excursions in the real-time index values. Annales Geophysicae, 36, 621–631. https://doi.org/10.5194/angeo-36-621-2018 .

Stauning, P. (2020). The Polar Cap (PC) index: invalid index series and a different approach". https://doi.org/10.1029/2020SW002442

Troshichev, O. A. and Janzhura, A. S. (2012). Physical implications of discrepancy between summer and winter PC indices observed in the course of magnetospheric substorms. Advances in Space Research, 50 (1), 77-84. https://doi.org/10.1016/j.asr.2012.03.017

Additional documentation available in Suppl.References.zip files :

Document received 25 February 2013 appended application for IAGA endorsement of PC index versions: PC_index_description_main_document_incl_Appendix_A.pdf (Matzka, 2014)

IAGA document on IAGA Task Force recommendation: PC_Task_Force_Recommendation_IAGA_2013.pdf

Letter from 18 May, 2018 from Professor M. Mandea to P. Stauning on behalf of IAGA Executive Committee and Division Leaders

Please also note the supplement to this comment: https://angeo.copernicus.org/preprints/angeo-2020-53/angeo-2020-53-AC3-supplement.zip

———————————————————

[Figure]

**Abstract.** A method is proposed to determine in near-real time the interplanetary magnetic field (IMF) sector structure (SS) effect on geomagnetic data for polar cap stations. To separate the SS effect, whose polarity is invariant within an interval from some days to 2 weeks, from the disturbed solar wind effects with periodicity from minutes to hours, the daily median values of geomagnetic H (or D) component are estimated. Then the median values for 9 days preceding the current day are subjected to 3-days running averages and the interpolation procedure is applied to these smoothed averages.

The proposed simple method makes possible identification of the SS effect in the same near real-time regime as the derivation of the quiet daily curve and as level of reference for the polar cap magnetic disturbances in the calculation of the polar cap magnetic activity PC index.

**The essential Fig. 6 of J&T2011**

Keeping in mind this specification, the 3-day smoothing averages of the median values were subjected to the interpolation procedure including the following steps:

1. median values for magnetic components H and D are derived for 4 intervals of preceding days with the exception of the current day ($n = 0$):

   - r1 = F [for interval from $n - 3$ day to $n - 1$ day]
   - r2 = F [for interval from $n - 5$ day to $n - 3$ day]
   - r3 = F [for interval from $n - 7$ day to $n - 5$ day]
   - r4 = F [for interval from $n - 9$ day to $n - 7$ day];

2. piecewise polynomial form of the cubic spline interpolant for r1, r2, r3, and r4 segments is determined;

3. termination of this form related to day $n = 0$ is examined as representative of the SS effect for the current day $n = 0$, even if this day is disturbed.

The procedure is repeated each subsequent day. Results of the procedure, the variation of the reconstructed magnetic H-component, are presented by the magenta line in the same Fig. 6, the reconstructed H-component curve being shifted by 50 nT to a lower position. Comparison of the red and magenta lines shows their good consistency in phase with the SS effect and quite satisfactory consistency in the amplitude of the SS effect.

[Figure]

**Fig. 6.** Behavior of the median values of the magnetic H-component at Thule station during June months of 1998 **(a)** and 2001 **(b)** for intervals with duration of 1 day (blue line), 3 days (green line), and 5 days (black line). The red dotted line shows the variation of the IMF $B_y$ component, derived from spacecraft measurements. The magenta line shows the variation of the reconstructed magnetic H-component. To be clearly demonstrated, the actual $B_y$ values were multiplied by five and were shifted by 50 nT to a higher position, whereas the curve of reconstructed H-component was shifted by 50 nT to a lower position.

*Acknowledgements.* Topical Editor R. Nakamura thanks B. Emery and C. Meng for their help in evaluating this paper.

**Fig. 1.** Essential features of Janzhura and Troshichev (2011): "Identification of the IMF sector structure in near-real time by ground magnetic data".

[Figure]

THL - H  3 days medians     Solar sector terms      Date: 30 Jun 2001

**Fig. 2.** The Hss values from Fig. 6b of J&T2011 are shown by the smooth line on the scale to the right, while the Hss values calculated by Cubic Spline extrapolation are shown by the dashed magenta line.

---

## Referee Comment (RC1) · Anonymous Referee #1 · 9 Feb 2021

This manuscript comment [Stauning 2020] addresses a paper by A. S. Janzhura and O. A. Troshichev titled "Identification of the IMF sector structure in near-real time by ground magnetic data" and published in Annales Geophysicae 9, 1491-1500, 2011 (subsequently referred to as [J & T 2011]).

As stated in the abstract of [J & T 2011]:

"A method is proposed to determine in near-real time the interplanetary magnetic field (IMF) sector structure (SS) effect on geomagnetic data from polar cap stations. [...] The

proposed simple method makes possible identification of the SS effect in the same near real-time regime as the derivation of the quiet daily curve and as level of reference for the polar cap magnetic disturbances in the calculation of the polar cap magnetic activity PC index."

The simple method described in section 3 uses 4 median values from the previous 9 days (n=1-3, 3-5, 5-7, 7-9) to fit a cubic spline and obtain a value for the current day (n=0). Although this is referred to as an "interpolation procedure", it is clearly extrapolation.

However, [Stauning 2020] asserts that several key results are actually obtained using post-event ("final") values based on daily medians smoothed over a 7-day interval centered on the day of interest. By definition this approach could not be applied in real-time. This would mean that [J & T 2011] did not prove the validity of their approach.

It should be noted that results in [J & T 2011] do show a clear connection between IMF sector structure and post-event ground magnetic data. They do not however directly address the usefulness of this connection for real-time monitoring as defined.

[Stauning 2020] addresses this issue by undertaking re-analysis based on extrapolation from the preceding 9-day intervals. Many results are significantly different in detail, although many general relationships are still evident. For example, linear correlation coefficients obtained by [Stauning 2020] from post-event analysis were in good agreement (r $\sim$ 0.95) with the results presented by [J & T 2011]. In comparison, re-analysis with the "real-time" algorithm produced much lower correlations (r $\sim$ 0.7).

If [Stauning 2020] has in fact discovered a significant defect in a paper with more than a dozen citations then some record of this should be published. A subsequent reply by the authors of [J & T 2011] would provide an opportunity to present additional re-analysis and determine how well their original thesis is supported in the light of new results.

---

## Author Comment (AC4) · 9 Feb 2021

Reply to RC1: (https://doi.org/10.5194/angeo-2020-53-RC1 )

Referee #1 has expressed very precisely my concerns over the publication Janzhura and Troshichev (2011) which has Dr. Troshichev as the corresponding author. Having detected by recalculations adhering meticulously to authors' definition of their "real-time" method that the illustrations were different from the ones presented in their Figs. 1, 6, 7, and 8, I completely concur with reviewer's conclusion that "some record of this

should be published".

Reply to SC1: (https://doi.org/10.5194/angeo-2020-53-SC1 )

Detailed point-by-point replies to the SC1 submitted by Dr. Troshichev have been provided in my former interactive reply (https://doi.org/10.5194/angeo-2020-53-AC1 ). To this reply I would like to add the final comment from Referee#1 that upon publication of my comment: "A subsequent reply by the authors of [J & T 2011] would provide an opportunity to present additional reanalysis and determine how well their original thesis is supported in the light of new results". C2

---

## Author Response (AR1)

Copenhagen 3 March 2021/PSt

**Responses to AnGeo Interactive Reviews on:**

Manuscript https://doi.org/10.5192/angeo-2020-53 "Comment on "Identification of the IMF sector structure in near-real time by ground magnetic data" by Janzhura and Troshichev (2011).
" by Peter Stauning.

I most gratefully acknowledge the efforts invested in the report from an anonymous referee (https://doi.org/10.5194/angeo-2020-53-RC1) as well as the short comment (https://doi.org/10.5194/angeo-2020-53-SC1 from Dr. Troshichev, (corresponding) author of the commented article.

**Handling of interactive comments**
The specific comments from anonymous Referee #1 have been addressed in Author Comment AC4, https://doi.org/10.5194/angeo-2020-53-AC4 . I completely agree with reviewer's points which are well represented in the manuscript.

The comments from Dr. Troshichev, corresponding author of the commented article, have been addressed in Author Comment AC1, https://doi.org/10.5194/angeo-2020-53-AC1 .
A specific point raised by Dr. Troshichev deals with the IAGA endorsement procedure. Dr. Troshichev mentions the work by a special Task Force team up a V-DAT meeting in May 2010 as basis for the endorsement of the PC index. However, the IAGA Task Force comprising Drs. M. Menvielle, H. McCreadie and C. Demetrescu delivered their report "PC Index – IAGA Endorsement" in the document: IAGA_documentation_20130225.pdf dated 20 Aug. 2013.
This document does not mention a V-DAT meeting in May 2010.
Supplementary comments on the publication by Janzhura and Troshichev (2011) and a summary of the IAGA PC endorsement handling are provided in https://doi.org/10.5194/angeo-2020-53-AC2 and https://doi.org/10.5194/angeo-2020-53-AC3

**Changes in the submitted manuscript**
Following the initial review the modified version dated 8 November 2020 was accepted for preview at the AnGeo Discussion Portal as https://doi.org/10.5194/angeo-2020-53
Compared to the 8 November 2020 version, the text in the present "Final Manuscript" dated 3 March 2021 has been changed by updating the reference to DTU Space portal in two places from ftp://ftp.space.dtu.dk/WDC/indices/pcn/ to https://doi.org/10.11581/DTU:00000057
The figure parts presented in separate images in the 8 November 2020 version have been combined to be presented in one figure for each set. The figures have not been modified otherwise.

Copenhagen 3 March 2021

Peter Stauning
pst@dmi.dk